

# First-order transition in a model of prestige bias

**Brian Skinner**

Department of Physics, Ohio State University, Columbus, OH 43210, USA

## Abstract

One of the major benefits of belonging to a prestigious group is that it affects the way you are viewed by others. Here I use a simple mathematical model to explore the implications of this "prestige bias" when candidates undergo repeated rounds of evaluation. In the model, candidates who are evaluated most highly are admitted to a "prestige class", and their membership biases future rounds of evaluation in their favor. I use the language of Bayesian inference to describe this bias, and show that it can lead to a runaway effect in which the weight given to the prior expectation associated with a candidate's class becomes stronger with each round. Most dramatically, the strength of the prestige bias after many rounds undergoes a first-order transition as a function of the precision of the examination on which the evaluation is based.

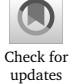

## 1 Introduction

It would seem that in just about any competitive human endeavor, there inevitably arises a notion of prestige. For example, students compete to get into prestigious universities, job-

seekers vie for positions at prestigious companies, researchers jockey to publish in prestigious journals and dream of winning prestigious awards, etc. But if you spend enough time in the treadmill of any of these competitive industries, you naturally end up asking the question: what is the utility of all this prestige? Why do we need a "prestige class"? Does it provide our industry with some benefit? And how does our focus on prestige alter the functioning of the industry?

Motivated by these kinds of questions, in this paper I focus on describing mathematically the way in which the notion of prestige affects our processes of selection and evaluation. The primary viewpoint adopted here is that a major function of prestige is to affect the way a person is evaluated by others. If a particular person is a member of the prestige class, then their candidacy during selection and evaluation processes is given a boost relative to similar candidates that do not belong to the prestige class. For example, two applicants for graduate school may have similar grades and exam scores, but if one candidate comes from a more prestigious university then their application will, in general, be evaluated more highly. This "prestige bias" arises naturally, since metrics like grades and exam scores are imprecise measures of a student's ability, and thus the evaluator looks for any other information available to help with their decision. Belonging to a prestigious group suggests that the candidate was ranked highly by some other evaluator in the past, and this provides a prior expectation (like a second opinion) that biases the decision in their favor.

This dynamic, in which a candidate to whom prestige has been conferred enjoys significant advantages over other candidates, has been the subject of a significant amount of quantitative research, particularly as it relates to academic hiring and evaluation. For example, it has long been recognized that academic science exhibits a "rich get richer" effect [1], wherein prestige grants opportunity and opportunity enables greater prestige. More recent work has focused on quantifying "prestige hierarchies" among academic institutions [2], and shown that the prestige ranking of a researcher's PhD institution is a very strong predictor of their success on the academic job market. Further, this institutional prestige largely predicts the productivity of early-career faculty [3], and seems to heavily influence which research topics develop a strong following [4]. Prestige also biases the way that academic papers are reviewed. For example, in a recent experiment with two disjoint program committees for an academic conference, the committee with access to the authors' identities and institutions was between 1.6 and 2.1 times more likely to accept papers from famous authors and/or prestigious institutions relative to the committee for which such information was blinded [5]. (See Ref. [6] for a larger review of this effect.) Prestige bias can have a particularly pernicious interaction with demographic bias [7], and much work has been devoted to studying the driving factors for the under-representation of women, in particular, in academic positions (see, e.g., Refs. [8–11]).

In this paper, the major goal is to suggest a model for thinking about prestige bias using the language of Bayesian inference across iterated rounds of evaluation. Similar language has been used to discuss "herd behavior" in decision-making processes like investing [12], in the review of academic papers [13], and in the adoption of academic paradigms [14]. In terms of the evaluation of candidates, Bayesian inference provides a way to incorporate information from an imprecise examination of the candidate with prior knowledge about the class from which the candidate comes. This process, when implemented optimally, maximizes the probability for the evaluator to select the best candidate. But it also provides a strong advantage for candidates from the prestige class, and can lead to persistent biases.

The model considered here is of a repeated process of examination and evaluation, in which evaluators use their knowledge of a candidate's class and their understanding of the distribution of abilities among members of that class to assess and rank candidates. After each round of evaluation a new "prestige class" is defined, which forms the basis for a Bayesian prior for the next round of evaluations, and so on. I study the dynamics of this selection process

and the degree of advantage it provides for members of the prestige class.

Crucial to the behavior of the model is the question of how the prior expectation of candidate ability is formed in the mind of the evaluator. I consider two different cases. In Case I, the prior expectation during each round is a perfectly accurate reflection of the actual range of abilities of each class. In Case II, the prior is constructed based on the estimates in ability from the previous round of evaluation. Case II exhibits qualitatively different dynamics in the evaluation process as compared to Case I, and in particular it can produce a runaway effect in which the weight given to the prior expectation becomes stronger with each successive round of evaluation.

Whether such a "runaway prestige bias" scenario occurs in Case II turns out to depend on the precision of the examination to which candidates are subjected. The most dramatic feature of the model is that, as a function of decreasing exam precision, there is a first-order transition in the weight given to the prior expectation after many rounds of evaluation. When the exam is precise, it allows evaluators to form accurate assessments of each candidate, rather than basing their assessment strongly on the prior expectation for the candidate's class. When the exam is less precise than this critical value, however, there is a runaway prestige bias effect that is accompanied by a "freezing out" of the prestige class, such that the prestige class becomes impossible to break into or fall out of.

In the remainder of this paper, I first define the model being studied (Sec. 2), and then give a simplified derivation of the first-order transition that produces the runaway prestige bias effect (Sec. 3). I then provide numerical results in Sec. 4 and conclude with a brief discussion in Sec. 5.

## 2  Model

In the model considered here, a large number $N$ of candidates undergo a process of repeated evaluation. For simplicity, I assume that candidates are characterized by only a single parameter $x$, which I refer to as "ability", that is constant and unchanging in time. For definiteness one can take $x$ to be distributed among the candidates according to a normal (Gaussian) distribution with mean 0 and variance 1 – this choice defines the units for ability. In the future, a normal distribution of the variable $x$ with mean $\mu$ and variance $\sigma^2$ will be denoted $\mathcal{N}(x; \mu, \sigma^2)$, so that one can write

$$\text{Prob}(x) = \mathcal{N}(x; \mu = 0, \sigma^2 = 1). \tag{1}$$

The goal of the evaluation process is to identify those candidates with the highest ability. Each round of evaluation begins with all candidates taking an *exam* (one could equally well decide to call it an "interview"). Let us suppose that the exam score is an unbiased estimator of the ability $x$ of the candidate, with a finite, constant precision. (This is a somewhat optimistic assumption, and the effects of systematic biases in the exam are discussed in Sec. 5.) In particular, let's say that for a candidate with ability $x$ the exam score $s$ is chosen from a normal distribution with mean $x$ and standard deviation $w$. In other words, the probability density for a candidate with ability $x$ to get a score $s$ on the exam is

$$\text{Prob}(s|x) = \mathcal{N}(s; \mu = x, \sigma^2 = w^2). \tag{2}$$

I refer to $w$ as the standard error of the exam; larger $w$ implies a less precise exam. (The quantity $1/w^2$ is often called the "precision".) $w$ is the one major parameter of the model.

After each round of exams, the candidates are given a ranking that depends on their exam score. In the first round of evaluation, evaluators have no prior expectation about each candidate, and so candidates are ranked simply in descending order by their exam score $s$. The

top-scoring fraction $f$ are admitted into the "prestige class", while the remaining $1 - f$ are considered to be part of the "non-prestige class". The parameter $f$ can be taken as a measure of the exclusivity of the prestige class. For definiteness I fix $f = 0.05$ for all numerical calculations in this paper.[1]

In the second and all subsequent rounds of evaluation, the evaluator knows both the candidate's exam score $s$ and whether the candidate belongs to the prestige class. This latter piece of information brings with it a prior belief about the candidate's likely ability, encoded in the form of a prior distribution $P_{\text{prior}}(x)$. Bayesian inference gives the optimal way to combine the exam score $s$ with the prior $P_{\text{prior}}(x)$, by using Bayes' Rule: [15]

$$\text{Prob}(x|s) = \frac{\text{Prob}(s|x)}{\text{Prob}(s)} P_{\text{prior}}(x). \tag{3}$$

Here, $\text{Prob}(s|x)$ is the probability for a candidate with ability $x$ to achieve the score $s$ on the exam; in this model it is given by Eq. (2). $\text{Prob}(s) = \int \text{Prob}(s|x) P_{\text{prior}}(x) dx$ is a normalization factor. From Eq. (3) one can calculate the expectation value of the candidate's ability, $\int x \text{Prob}(x|s) dx$. I will refer to this expectation value as the *estimated ability* $\varepsilon(s)$ of a candidate with exam score $s$. In particular,

$$\varepsilon(s) = \frac{\int x \mathcal{N}(s; \mu = x, \sigma^2 = w^2) P_{\text{prior}}(x) dx}{\int \mathcal{N}(s; \mu = x, \sigma^2 = w^2) P_{\text{prior}}(x) dx}. \tag{4}$$

In the special case where the prior distribution is also normal, $P_{\text{prior}}(x) = \mathcal{N}(x; \mu = \mu_{\text{prior}}, \sigma^2 = \sigma^2_{\text{prior}})$, this equation can be evaluated explicitly, giving [15]

$$\varepsilon(s) = \frac{\mu_{\text{prior}} w^2 + s \sigma^2_{\text{prior}}}{w^2 + \sigma^2_{\text{prior}}}. \tag{5}$$

Notice that $\varepsilon(s) \to s$ in the limit where the exam is very precise compared to the prior ($w \ll \sigma_{\text{prior}}$), and $\varepsilon(s) \to \mu_{\text{prior}}$ in the opposite limit of high standard error ($w \gg \sigma_{\text{prior}}$). In the remainder of this paper I will work within the assumption of a Gaussian prior, so that Eq. (5) is applicable. In other words, the prior distribution used by evaluators is taken to be characterized fully by its mean $\mu_{\text{prior}}$ and variance $\sigma^2_{\text{prior}}$.

The crucial question, and the central topic of this paper, is how the prior distribution is formed in the mind of the evaluator. In general, the prior for a given round of evaluation is based on the abilities, or perceived abilities, of the class to which the candidate belongs (prestige or non-prestige). One can consider two extreme cases for how an evaluator might form their perception of the candidate's class:

- *Case I: Prior based on perfect knowledge.* The most idealistic scenario is that the evaluator knows with perfect accuracy the distribution of abilities within each class. That is, following each round of evaluation, the evaluator becomes well acquainted with the members of each class (or, at least, with a representative set of members from each class), and comes to understand their true abilities. This knowledge allows the evaluator to form a perfectly accurate estimation of the mean and variance in ability for the candidate's class, which can be used in the next round of evaluations.

  Thus, in Case I, $\mu_{\text{prior}}$ is equal to the mean in true ability $\mu_x$ for the candidate's class, and $\sigma^2_{\text{prior}}$ is equal to the variance in true ability $\sigma^2_x$.

---

[1] In principle, $f$ is a second parameter of the model, which can be explored on equal footing with $w$. In practice, however, the quantities of interest have a very weak dependence on $f$ when $f \ll 1$; see Appendix A.

- *Case II: Prior based on past evaluation.* A less optimistic (but, unfortunately, probably more realistic) scenario is one in which the evaluators acquire no new knowledge about the candidates after the evaluation. In this sense, the evaluators do not have enough information to form an accurate prior. Still, the evaluators may conclude that they *do* have knowledge of the candidates' abilities, since they explicitly estimated their abilities during the previous round of evaluation. For such evaluators, the natural choice of a prior distribution is the distribution of estimated abilities $\varepsilon$ from the previous round.

  In other words, in Case II, $\mu_{\text{prior}}$ for round $n$ is equal to the mean in estimated ability $\mu_\varepsilon$ from the previous round $n-1$, and similarly $\sigma^2_{\text{prior}} = \sigma^2_\varepsilon$.

In principle, one can interpolate continuously between Case I and Case II. That is, an evaluator may form their opinion about the abilities of each class by combining their past evaluations with some new information, so that the mean and variance in the prior distribution are intermediate between $(\mu_x, \sigma^2_x)$ and $(\mu_\varepsilon, \sigma^2_\varepsilon)$. This kind of interpolation would introduce another parameter into the model and is not considered here, outside of a brief comment in Sec. 5.

After each round of evaluation, candidates are ranked according to their estimated ability $\varepsilon$, and the top fraction $f$ comprise the prestige class during the subsequent round.

## 3 Theory of the first-order transition

In this section I use a simplified theoretical description to derive the primary feature of this model, namely the first-order transition as a function of the exam precision that occurs in Case II. The major simplification used in this section is to assume that all statistical distributions are normal. As will be shown by numeric calculations in the following section, the relevant distributions are not normal in general, and this may alter certain numerical values but not the qualitative features being derived.

To begin, let us consider the relationship between three statistical distributions that characterize a particular class (prestige or non-prestige). The first is the distribution of abilities, $x$, for members within the class. The second is the distribution of the class's exam scores $s$. The third is the distribution of estimated ability $\varepsilon$, as perceived by the evaluators. Let us suppose that the distribution of abilities among members of a given class is characterized by a mean $\mu_x$ and a variance $\sigma^2_x$. When these candidates take the exam, the distribution of their exam scores can be found by convolving the distribution $\text{Prob}(x)$ with $\text{Prob}(s|x)$. This process gives a mean score $\mu_s = \mu_x$ (since the exam is unbiased), while the variance in exam scores is somewhat larger than the variance in ability, $\sigma^2_s = \sigma^2_x + w^2$.

The process of Bayesian estimation, however, reduces the variance in the exam score by effectively averaging the exam score with the prior mean. One can see this reduction by, for example, rearranging Eq. (5) as

$$\varepsilon(s) = \mu_{\text{prior}} + (s - \mu_{\text{prior}}) \frac{\sigma^2_{\text{prior}}}{w^2 + \sigma^2_{\text{prior}}}.$$

The second term in this equation shows that the range of exam scores among candidates is mapped onto a narrower range of estimated ability, with a reduction in range by a factor $\sigma^2_{\text{prior}}/(w^2 + \sigma^2_{\text{prior}})$. In essence, through the process of Bayesian inference an evaluator "corrects" an unusually large or small exam score to bring it closer in line with the prior expectation for the group mean. When the standard error of the exam is much wider than the width of the prior, $w^2 \gg \sigma^2_{\text{prior}}$, the candidate is generally assumed to have an ability very close to that of the prior mean $\mu_{\text{prior}}$, regardless of the candidate's score on the exam.

The resulting variance in estimated ability, $\sigma_\varepsilon^2$, can be calculated using $\sigma_\varepsilon^2 = \int [\varepsilon(s) - \mu_\varepsilon]^2 \mathrm{Prob}(s) ds$, which gives

$$\sigma_\varepsilon^2 = \frac{\sigma_{\mathrm{prior}}^4 (w^2 + \sigma_x^2)}{(w^2 + \sigma_{\mathrm{prior}}^2)^2}. \tag{6}$$

This equation implies, in general, a reduction in the variance of $\varepsilon$ relative to the variance $\sigma_x^2$ in actual ability. For example, even in the case when the prior is perfectly well-formed, such that $\sigma_{\mathrm{prior}}^2 = \sigma_x^2$, the variance in estimated ability $\sigma_\varepsilon^2$ is smaller than $\sigma_x^2$ by a factor $\sigma_\varepsilon^2/\sigma_x^2 = \sigma_x^2/(w^2 + \sigma_x^2) < 1$. Notice that this reduction is especially large when the exam is very imprecise, $w \gg \sigma_x^2$. In effect, when the exam is a sufficiently poor indicator of candidate ability that it cannot distinguish between different members of the same class, evaluators become conservative and tend to assume that all candidates are simply typical of their class.

The rate at which candidates are "promoted" to the prestige class, or "demoted" from it, depends on the overlap of the distributions of estimated ability $\varepsilon$ for the two classes. In this sense, the smallness of $\sigma_\varepsilon^2$ that arises from the evaluators' conservative estimation makes class mobility artificially small. In Case I, this issue of low mobility is at least not compounded from one round to another, since the prior expectation is accurately recalibrated after every round of evaluation. However, in Case II the smallness of $\sigma_\varepsilon^2$ arising from Bayesian inference can become a persistent problem, because it forms the basis for the prior used during the next round.

Let us now focus on the evolution of $\sigma_\varepsilon^2$ with $n$ in Case II. In this case, the prior distribution for evaluation during round $n+1$ is based on the distribution of estimated ability $\varepsilon$ from the previous round, and thus $\sigma_{\mathrm{prior},n+1}^2 = \sigma_{\varepsilon,n}^2$. Equation (6) therefore defines a recurrence relation for $\sigma_\varepsilon^2$:

$$\sigma_{\varepsilon,n+1}^2 = \sigma_{\varepsilon,n}^2 \frac{\sigma_{\varepsilon,n}^2 (\sigma_x^2 + w^2)}{(\sigma_{\varepsilon,n}^2 + w^2)^2}. \tag{7}$$

Let us treat $\sigma_x^2$ as a constant, independent of $n$. This assumption is not strictly correct, since the members of the class may change from one round to another. However, $\sigma_x^2$ cannot change parametrically with time, since its maximum value is $\sigma_x^2 = 1$ (corresponding to a randomly-chosen prestige class) and its minimum value for the chosen value of $f$ is $\sigma_x^2 \approx 0.4$ (a perfectly-selected prestige class; see Appendix A for a derivation). With this assumption of constant $\sigma_x^2$, one can notice that Eq. (7) has two nonzero fixed points, defined by $\sigma_{\varepsilon,n+1}^2 = \sigma_{\varepsilon,n}^2 \equiv \sigma_{\varepsilon,\mathrm{fixed}}^2$:

$$\sigma_{\varepsilon,\mathrm{fixed}}^2 = \frac{\sigma_x^2 - w^2}{2} \pm \frac{\sqrt{(\sigma_x^2 + w^2)(\sigma_x^2 - 3w^2)}}{2}. \tag{8}$$

The larger of these two fixed points is stable, in the sense that repeated rounds of evaluation tend to bring the value of $\sigma_{\varepsilon,n}^2$ closer to the larger fixed point, while the lesser of the two fixed points is unstable. The flow of $\sigma_{\varepsilon,n}^2$ with increased $n$ is illustrated in Fig. 1.

When the standard error $w$ of the exam exceeds a certain critical value $w_c$, the only remaining fixed point is $\sigma_\varepsilon^2 = 0$. In particular,

$$\sigma_{\varepsilon,\mathrm{fixed}}^2 = 0 \ \mathrm{at} \ w > w_c = \frac{\sigma_x}{\sqrt{3}}. \tag{9}$$

The vanishing of the nonzero fixed points at $w > w_c$ implies that there is a *first-order transition* in the fixed-point value of $\sigma_\varepsilon^2$ as a function of the standard error. That is, when $w < w_c$, the variance in estimated ability $\sigma_\varepsilon^2$ approaches a finite value after many rounds of evaluation (so long as the initial value of $\sigma_\varepsilon^2$ is not very small). However, when $w > w_c$, the value of $\sigma_\varepsilon^2 \to 0$ in the limit of large $n$.

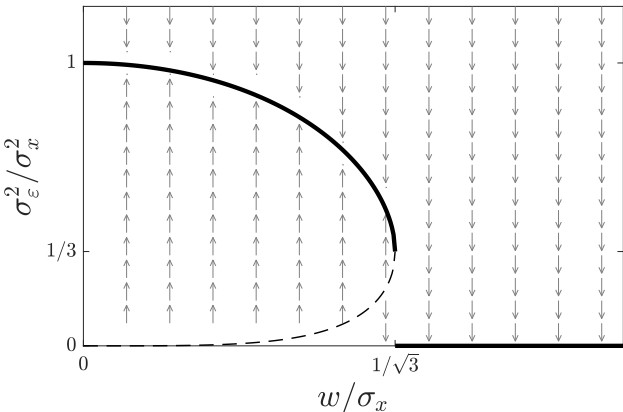

Figure 1: Flow of the variance in estimated ability, $\sigma_\varepsilon^2$, for a given class of candidates under repeated rounds of evaluation. $\sigma_\varepsilon^2$ determines the weight given to the prior expectation, such that when $\sigma_\varepsilon^2 \to 0$ the prior expectation in dominant and the candidate's exam has no influence on their assessed ability. The horizontal axis indicates the standard error of the exam, $w$. Both $\sigma_\varepsilon$ and $w$ are normalized to the standard deviation in actual candidate ability, $\sigma_x$. The thick, solid line shows the stable fixed point as a function of $w$, and arrows indicate the direction of flow under repeated rounds of evaluation. The dashed line shows the unstable fixed point. The discontinuity in the thick line indicates a discontinuous transition in the fixed point of $\sigma_\varepsilon^2$ as a function of $w$.

This transition in $\sigma_\varepsilon^2$ as a function of $w$ implies a transition in the way the evaluation process works at long times. As long as $\sigma_\varepsilon$ is finite, there remains a finite chance for a candidate in a given class to distinguish themselves (either positively or negatively) by achieving an exceptional score on the exam. However, when $\sigma_\varepsilon \to 0$, the prior becomes infinitely strong, and all members within the class are viewed identically. One can think of this collapse of $\sigma_\varepsilon$ as a "runaway confirmation bias" effect, in which the evaluator's impression about a particular class gets stronger and more specific with each round of evaluation. The existence of a $\sigma_\varepsilon \to 0$ transition for a given class does not depend on the interaction between the two classes.

In fact, since there are two classes, each with a different range of ability for its members, there are in general two transitions as a function of $w$. At sufficiently small $w$, the fixed-point value of $\sigma_\varepsilon^2$ is finite for both classes, and candidates are exchanged between the two classes at a relatively high rate even in the long-time limit. On the other hand, when $w$ is very large, both classes undergo the collapse of the prior illustrated in Fig. 1, and at long times all exchange between the classes is frozen out. However, since $\sigma_x^2$ is smaller for the prestige class than for the non-prestige class (provided that $f < 1/2$), there exists a range of $w$ for which the prestige class has a collapsed prior while the non-prestige class does not. In this range it remains possible for a member of the non-prestige class to score sufficiently highly on the exam to be promoted into the prestige class, but such exchanges occur with an exponentially low rate.

Thus, in terms of inter-class mobility, there are two transitions as a function of increasing $w$. At the first transition the rate of inter-class exchange collapses from an order-unity value to an exponentially small value, as the prior becomes infinitely strong for the prestige class but not for the non-prestige class. At the second transition $\sigma_\varepsilon^2 \to 0$ for the non-prestige class as well, and the rate of inter-class exchange collapses to zero.

# 4 Numerical results

In this section I present results from a numerical simulation of the dynamics defined in Sec. 2. I explore how the distributions of actual ability and estimated ability evolve over successive rounds of evaluation, as well as the rate of exchange between the two classes. Results are presented for Case I, where the prior is based on accurate knowledge of the two classes, and for Case II, where the prior is based on past evaluations.

Specifically, I simulate a large cohort of $N$ candidates, each with ability $x$ chosen according to Eq. (1). During each round, each candidate is assigned an exam score according to Eq. (2), and their estimated ability is evaluated according to Eq. (5). For Case I, the prior is chosen to reflect the true abilities of members of the candidate's class, and for Case II the prior reflects the estimates from the previous round of evaluations (as detailed in Sec. 2). After $\varepsilon_i$ is estimated for every candidate $i$, those with the top scores are sorted into the prestige class, and the process is repeated.

In addition to examining the statistical distributions of $x$ and $\varepsilon$, I also examine the rate at which candidates can expect to move from one class to another. In particular, I define the *class mobility*, $M$, as follows. Consider a candidate with ability that is precisely at the lower margin of an ideally-chosen elite class. That is, consider a candidate with ability $x_{\mathrm{m}} = \sqrt{2}\,\mathrm{erfc}^{-1}(2f)$, where $\mathrm{erfc}^{-1}(z)$ is the inverse complementary error function, so that $x_{\mathrm{m}}$ constitutes the $(1-f)$th percentile of all candidates. $M$ is defined as the probability for such a candidate to score highly enough on the exam to be promoted from the non-prestige to the prestige class. In situations with precise exams and weak priors, $M$ approaches $1/2$. On the other hand, when the prior is strong and the exam is imprecise, $M$ approaches zero. The transition in $\sigma_\varepsilon^2$ implies a concomitant collapse in the class mobility $M$.

Unless otherwise noted, all data below corresponds to simulations of $N = 10^5$ candidates and is averaged over 1000 realizations.

## 4.1 Case I: Prior based on perfect knowledge

In Case I, the distributions of actual and estimated ability quickly converge to a steady state under increased iteration $n$. See, for example, Fig. 2, which considers the case of $w = 0.3$; as shown below, this value of $w$ is large enough to induce a collapse in the prior distribution for Case II. But in Case I there is no such collapse, since the prior is accurately recalibrated at every round.

The lack of collapse in the prior distributions implies that in the steady state there is a reasonable amount of exchange between the two classes. This is illustrated in Fig. 3, which shows that the class mobility quickly saturates to a finite value with increasing $n$. The value of the mobility declines as the exam becomes less precise (i.e., when $w$ increases), since evaluators give more weight to the prior when the exam is imprecise. But in all cases the mobility remains finite in the steady state.

## 4.2 Case II: Prior based on past evaluation

Unlike in Case I, in Case II the recursive use of past evaluations can lead to a transition in the weight given to prior evaluations. This is shown explicitly in Fig. 4, which plots the variance in estimated ability of the prestige class as a function of the standard error $w$ of the exam. As predicted in Sec. 3, there is an abrupt transition in the value of $\sigma_\varepsilon^2$ at long times when $w$ exceeds a critical value. It is worth noting that the $y$-intercept in this plot gives the variance in ability of an ideally-selected prestige class, $\sigma_x^2 \approx 0.138$ (see also the derivation in Appendix A). The transition $\sigma_\varepsilon^2 \to 0$ occurs at $w = w_c \approx 0.215$, which is quantitatively consistent with the value $\sqrt{\sigma_x^2/3} \approx 0.214$ derived in Sec. 3.

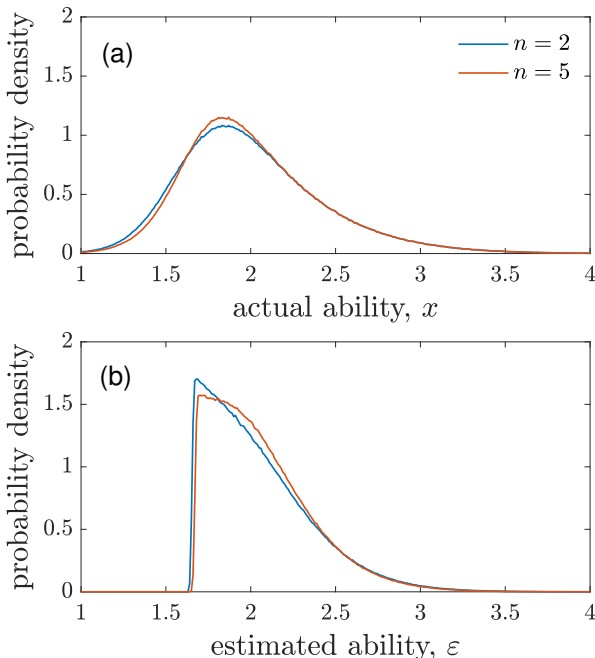

Figure 2: Distributions of actual ability (a) and estimated ability (b) for the prestige class under the dynamics of Case I, where the prior is based on accurate knowledge of the true abilities within each class. Only the distributions for $n = 2$ and $n = 5$ are shown; larger values of $n$ practically coincide with the latter. Here the standard error of the exam $w = 0.3$.

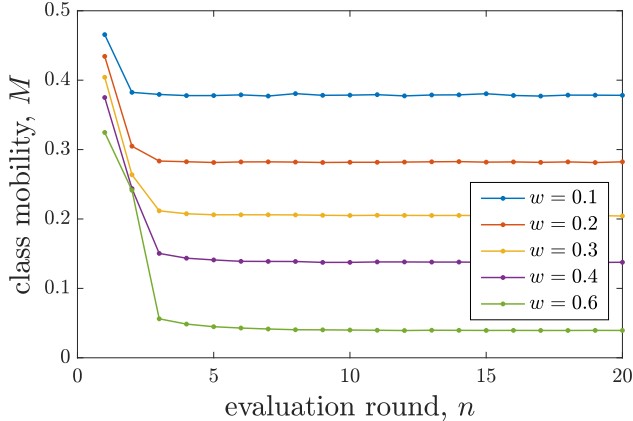

Figure 3: The class mobility $M$ as a function of time for Case I, plotted for different values of the standard error $w$ of the exam. $M$ quickly approaches a steady-state value, which declines with increasing $w$ but remains finite for all $w$ and $n$.

The effects of the transition can also be seen in the class mobility $M$ (Fig. 5). At $w < w_c$, the class mobility approaches a steady state value, similar to Case I (Fig. 3). However, when $w > w_c$ the class mobility abruptly collapses to a very small value, induced by the collapse of the prior for the prestige class. When $w$ is increased even further, such that $w > 1/\sqrt{3}$, the prior collapses for the non-prestige class as well, and $M \to 0$ at large $n$.

Finally, one can see the effects of the collapse in the distributions of $\varepsilon$. For example, Fig. 6 shows the distributions of $x$ and $\varepsilon$ at $w < w_c$, both of which are stable as a function of $n$ and have finite width. In contrast, at $w > w_c$ (Fig. 7), the distribution of estimated ability rapidly collapses to a delta function with increasing $n$.

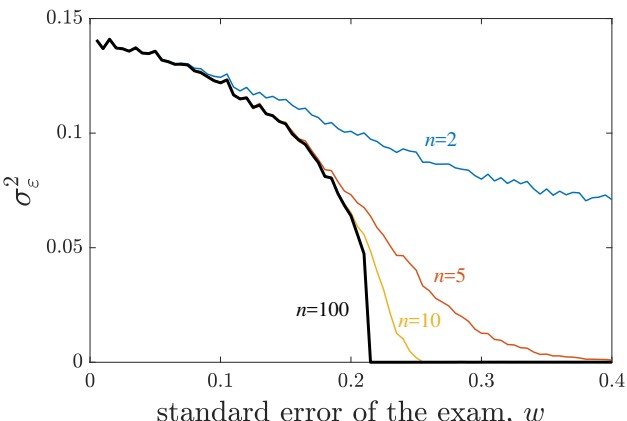

Figure 4: Variance $\sigma_\varepsilon^2$ in the estimated ability of the prestige class for Case II, plotted as a function of the standard error $w$ of the exam. Different curves correspond to different rounds $n$ of evaluation. Data is taken from numerical simulations of a single cohort of $N = 10^6$ candidates. Compare Fig. 1.

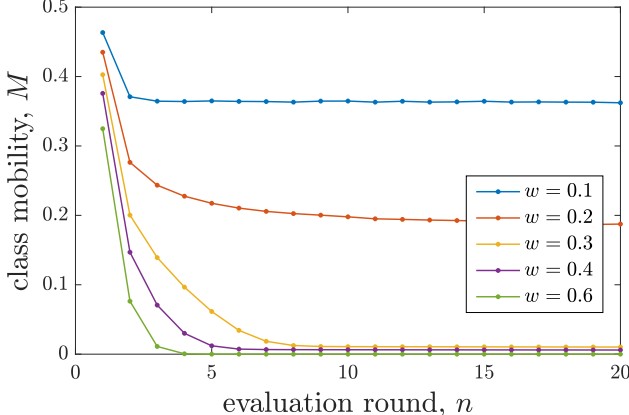

Figure 5: The class mobility $M$ as a function of time for Case II, plotted for different values of the standard error $w$ of the exam. When $w < w_c \approx 0.215$, $M$ acquires a reasonably large value in the steady state. When $w > w_c$, the prior for the prestige class collapses, and $M$ falls abruptly to a very small steady-state value. At $w > 1/\sqrt{3} \approx 0.577$ the prior collapses for the non-prestige class as well, and $M \to 0$.

## 5  Discussion

To summarize, this paper has presented a simple, single-parameter[2] model of prestige bias under a process of iterated evaluation. The major result is that there is a first-order transition in the way the evaluation operates after many rounds of evaluation. If the exam used for evaluation is less precise than a certain critical value, there is a "freezing out" of the prestige class, which blocks the turnover of candidates between classes regardless of their ability.

It is important to mention that the existence of prestige bias in this model does not require any assumptions about bad motivations or irrational behavior by the evaluators. Such bias arises naturally in a Bayesian description when rational actors undertake evaluations in the presence of incomplete information. The major message of this paper, however, is that evaluators must be conscientious about the basis for their prior assumptions about a candidate's

---

[2] In principle the exclusivity $f$ of the prestige class can be treated as an additional parameter, but see footnote 1 above and the comment at the end of Appendix A.

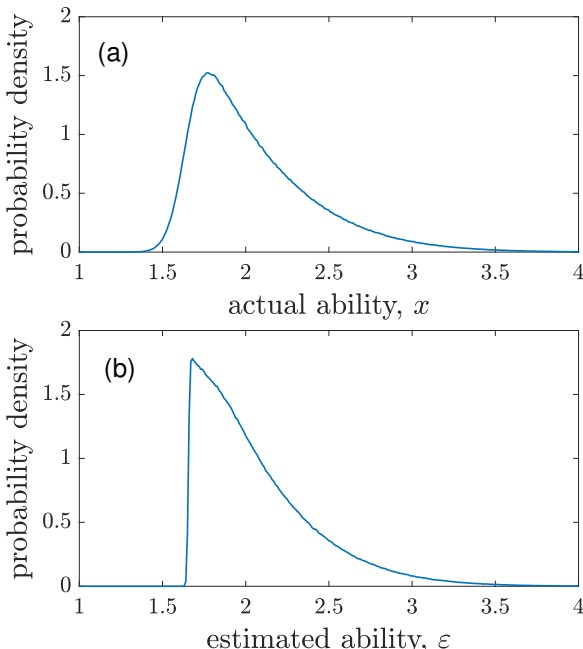

Figure 6: Distributions of actual (a) and estimated (b) ability for the prestige class under the dynamics of Case II, with $w = 0.1$. Here the standard error $w < w_c$, so that there is no collapse of the prior. Data corresponds to $n = 2$; curves for larger values of $n$ practically coincide.

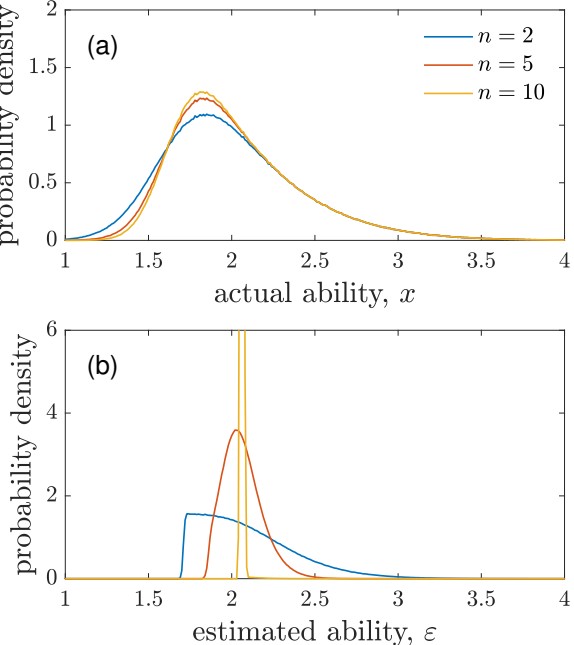

Figure 7: Distributions of actual (a) and estimated (b) ability for the prestige class under the dynamics of Case II, with $w = 0.3$. Here the standard error $w > w_c$, which causes the prior to collapse. This collapse is accompanied by a runaway narrowing in the distribution of estimated ability with increasing $n$, with a mean that is larger than the mean of actual ability.

class. A runaway bias effect occurs when evaluators heavily base their evaluations on their impressions from the previous round of evaluation. For example, a professor evaluating postdoc

applicants might think "ah, this candidate went to grad school at Elite University X, and I remember how impressive all the students from that university were when I was reviewing grad school admissions." This line of thinking is easy to fall into, but in effect it compounds one bias with another, artificially widening the advantage that students from Elite University X already experienced at the graduate school stage. A runaway prestige bias effect can be avoided only if there is a sufficiently accurate and unbiased examination of candidate abilities, or if the evaluator conscientiously resets their prior expectations at every stage of evaluation. This conscientious resetting would require, for example, a person evaluating graduate students from Elite University X to consider only their direct experiences with other graduate students at Elite University X, rather than drawing on a wider set of impressions that encompass individuals at other stages of their career.

Of course, the specific details of the model considered in this paper have been highly schematic, and one should be wary in general any time a physicist proposes a new "simple model" [16]. But, if one were to develop an interest in doing so, there are a number of ways that the model presented here could be improved or expanded upon. For example, the model assumes that candidate ability is static, and there is no feedback between a candidate's ability and their class. Incorporating such feedback may exacerbate the runaway effect being described. One could also imagine a model where "ability" is characterized by multiple parameters, rather than a single one, which could open up a wider space of possible phases and transitions. The dichotomy of Case I and Case II is also arbitrary, and, as mentioned briefly in Sec. 2, one could imagine introducing a parameter that interpolates between the two cases. For example, one could introduce an interpolation parameter $\beta$, such that the variance used in the prior satisfies $\sigma^2_{\text{prior},n+1} = \beta \sigma^2_x + (1-\beta)\sigma^2_{\varepsilon,n}\sigma^2$. In such a description, any $\beta > 0$ (i.e., any finite deviation from Case II) would truncate the transition and give a finite $\sigma^2_\varepsilon$ in the steady state.

Finally, it is worth making a comment about demographic bias. It was assumed in this paper that the exam is an unbiased estimator of candidate ability. But real-world examinations often have biases toward one demographic group or another, which lead to suboptimal ranking and selection of candidates (see, e.g., Refs. [17–19] for a few illustrative case studies). An especially interesting question suggested by the results here is whether bias that exists only in the early rounds of evaluation will persist indefinitely through later rounds. A natural conjecture is that the answer depends on whether $w < w_c$ or $w > w_c$. In the former case, any biases in the early rounds of exam eventually dissipate, since there remains a robust mobility between the two classes. In the latter case, however, the class mobility rapidly goes to zero, and any demographic biases from the early rounds are "frozen in." This question may be studied using relatively simple modifications of the model presented here, and deserves further exploration.

# Acknowledgments

The author acknowledges the various instances of demographic privilege and blind luck that have allowed his academic career to continue for as long as it has.

This work is unfunded.

## A  Mean and variance in ability for a perfectly-selected prestige class

Within the assumption of a normal distribution of candidate ability [Eq. (1)], an ideally-selected prestige class comprises a truncated normal distribution:

$$P_{\text{tr}}(x) = A \exp[-x^2/2]\Theta(x - x_{\text{m}}), \tag{10}$$

where $\Theta(x)$ is the Heaviside step function and $A$ is a normalization constant given by

$$A = \left[\int_{x_{\text{m}}}^{\infty} \exp[-x^2/2]dx\right]^{-1} = \frac{\sqrt{2/\pi}}{\text{erfc}[x_{\text{m}}/\sqrt{2}]}. \tag{11}$$

The cutoff $x_{\text{m}}$ can be calculated from the requirement that the total population of the prestige class comprises a fraction $f$ of all candidates: $\int_{x_{\text{m}}}^{\infty} \mathcal{N}(x; \mu = 1; \sigma^2 = 1)dx = f$. This relation gives

$$x_{\text{m}} = \sqrt{2}\,\text{erfc}^{-1}(2f), \tag{12}$$

as mentioned in Sec. 4. The mean ability of the optimally-selected prestige class is therefore

$$\mu_x = \int_{x_{\text{m}}}^{\infty} x P_{\text{tr}}(x)dx = \frac{1}{\sqrt{2\pi f^2}}\exp\left\{-\left[\text{erfc}^{-1}(2f)\right]^2\right\}, \tag{13}$$

and the variance is

$$\begin{aligned}
\sigma_x^2 &= \int_{x_{\text{m}}}^{\infty} (x - \mu_x)^2 P_{\text{tr}}(x)dx \\
&= 1 + \frac{\exp\left\{-\left[\text{erfc}^{-1}(2f)\right]^2\right\}\text{erfc}^{-1}(2f)}{\sqrt{\pi}f} \\
&\quad - \frac{\exp\left\{-2\left[\text{erfc}^{-1}(2f)\right]^2\right\}}{2\pi f^2}.
\end{aligned} \tag{14}$$

For $f = 0.05$, these expressions give $x_{\text{m}} = 1.645$, $\mu_x = 2.063$, and $\sigma_x^2 = 0.138$.

In the limit of asymptotically small $f$, Eqs. (13) and (14) become

$$\mu_x \simeq \sqrt{\ln\left(\frac{1}{2\pi f^2}\right)}, \tag{15}$$

and

$$\sigma_x^2 \simeq \left[\ln\left(\frac{1}{2\pi f^2}\right)\ln\left(\frac{1}{2\pi f^2 \ln(1/2\pi f^2)}\right)\right]^{-1/2}. \tag{16}$$

This extremely weak dependence on the exclusivity $f$ of the prestige class implies that, although in principle the first-order transition exists in a space of two parameters $w$ and $f$, in practice the value of $f$ may matter very little.

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
