# Peer review of "First-order transition in a model of prestige bias"

_SciPost Physics, doi:SciPost Phys. 8, 030 (2020)_

## Round 2 · Referee Report · Anonymous (Referee 1) · 2019-12-9

Strengths
-
This simple agent-based model shows how the interplay between prior bias and Bayesian updates lead to either no estimation error of the quality of people, or are not able to remove the influence of prior bias.
-
This submission is generally written in a clear way and leaves food for thought.
Weaknesses
- That said, I regret the wording "precision" for p: a high precision means a small p, which is very confusing, especially since it is the crucial parameter of the model, and this leads to unnecessary convoluted mental process. The paper proposes to think as power as the inverse of precision. Why not using power instead in the discussion, then?
Report
Requested changes
precision -> power ?

Anonymous on 2019-12-09 [id 670]
This simple agent-based model shows how the interplay between prior bias and Bayesian updates lead to either no estimation error of the quality of people, or are not able to remove the influence of prior bias.
This submission is generally written in a clear way and leaves food for thought.
That said, I regret the wording "precision" for p: a high precision means a small p, which is very confusing, especially since it is the crucial parameter of the model, and this leads to unnecessary convoluted mental process. The paper proposes to think as power as the inverse of precision. Why not using power instead in the discussion, then?
Last question: is there an intuitive reason for the fact that $p_c=1/\sqrt{3}$?

---

## Round 3 · Author Response

In the previous round of review, the referee expressed concern about referring to the variable $p$ as the "precision". "Precision" has a specific meaning in statistics, and refers to the inverse square of the variance, while in this model larger p means that the exam is _less_ precise. So referring to p as "precision" is likely to confuse readers.

I have corrected this bad terminology by replacing $p$ with $w$ and referring to this parameter as the "standard error of the exam." This term should be unambiguous. (The term "power", suggested by the referee, also has a specific meaning in statistics that is not exactly the same as the standard error of the exam, so I have avoided using it.)

Regarding the result $p_c = 1/\sqrt{3}$, unfortunately I don't have a more "intuitive" derivation than the one given in Section III of the text.

---

## Round 3 · List of Changes

- I replaced $w$ with $p$ and the term "precision" with "standard error".
- I corrected a couple small typos.

---

## Editorial Decision

published